# Microenvironmental Impact on InP/ZnS-Based Quantum Dots in In Vitro Models and in Living Cells: Spectrally- and Time-Resolved Luminescence Analysis

**DOI:** 10.3390/ijms24032699

**Published:** 2023-01-31

**Authors:** Ilia Litvinov, Anna Salova, Nikolay Aksenov, Elena Kornilova, Tatiana Belyaeva

**Affiliations:** 1Institute of Cytology, Russian Academy of Sciences,194064 Saint Petersburg, Russia; 2Higher School of Biomedical Systems and Technologies, Peter the Great St. Petersburg Polytechnic University, 195251 Saint Petersburg, Russia

**Keywords:** quantum dots, quantum yield, photoluminescence

## Abstract

Quantum dots (QDs) have attracted great attention as tools for theranostics that combine the possibility of simultaneous biological target visualization and medicine delivery. Here, we address whether core/shell InP/ZnS QDs (InP-QDs) may be an alternative to toxic Cd-based QDs. We analyze InP-QD photophysical characteristics in cell culture medium, salt solutions, and directly in the cells. It was demonstrated that InP-QDs were internalized into endolysosomes in HeLa and A549 cells with dynamics similar to Cd-based QDs of the same design, but the two cell lines accumulated them with different efficiencies. InP-QDs were reliably detected in the endosomes despite their low quantum yields. Cell culture medium efficiently decreased the InP-QD photoluminescence lifetime by 50%, acidic pH (4.0) had a moderate effect (20–25% reduction), and quenching by salt solutions typical of intra-endosomal medium composition resulted in a decrease of about 10–15%. The single-vesicle fluorescence-lifetime imaging microscopy analysis of QDs inside and outside the cells shows that the scatter between endosomes in the same cell can be significant, which indicates the complex impact of the abovementioned factors on the state of InP-QDs. The PI test and MTT test demonstrate that InP-QDs are toxic for both cell lines at concentrations higher than 20 nM. Possible reasons for InP-QD toxicity are discussed.

## 1. Introduction

Semiconductor nanocrystals, known as quantum dots (QDs), have unique spectral and luminescent properties, such as emission wavelengths covering the visible and near-infrared range of spectra depending on the chemical composition of the core and its size, large absorption coefficients in the UV range, narrow luminescence spectra, high brightness, and photostability [1,2]. Usually, QD consists of a core covered by a layer of ZnS to reduce nonradiative recombination, thus obtaining core/shell structures with higher quantum yield. Then, to turn hydrophobic nanocrystals into water-soluble form, QDs are encapsulated into another shell made of amphipatic molecules, more often polymeric. The relatively large surface of such functionalized nanocrystals with active groups (-${\mathrm{NH}}_{2}$, -COOH or -OH) makes it possible to attach either targeting molecules, such as ligands for cell surface receptors and antibodies, or various active substances (medicaments, cell-killing drugs, or photosensitizers), which makes QD-based systems attractive tools for imaging, both in basic research and theranostics. Currently, new fields of QD applications are being developed; an interesting example is the use of QDs as photocatalysts for the polymerization of monomers of different origins. In the context of biomedicine, it is particularly interesting to suggest that complexes with QDs could be created in this way to obtain targeted nanoparticles [3,4,5].

However, the widespread introduction of QDs into practice is associated with a number of problems. As CdSe-based QDs currently have the most advanced synthesis protocol, they are the most widely studied. Due to the risk of leakage of cadmium ions into the cytoplasm with subsequent attachment to various intracellular proteins through sulfhydryl groups [6], which causes a cytotoxic effect [7], the use of CdSe-QDs is possible but may be limited in biology and biomedicine [8,9]. The problem of this cytotoxicity is solved by reducing the concentration of QDs used for detection, strengthening the shell of QDs, and introducing QDs based on less toxic materials (elements) in practice [10,11]. The designs of non-metallic QDs are considered, for example, those based on carbon, silicon, or II-VI and III-V groups of elements [12]. As a promising solution to the problem of cytotoxicity, cadmium-free QDs based on InP have been proposed [13,14,15]. However, until now, there have been few detailed studies on the characterization of such QDs and their interactions with cells.

Concerning QD–cell interactions, it should be noted that QDs, after functionalization, reach sizes that allow them to enter cells only via endocytosis and are thus delivered into endosomes. The endosomal membrane contains a wide range of ion channels that enable rapid and dramatic changes in the concentration of the main ions (e.g., $K^{+}$, ${\mathrm{Na}}^{+}$, ${\mathrm{Ca}}^{2+}$, ${\mathrm{Cl}}^{-}$) inside the vesicle. In addition, endosomes contain ATPase, which pumps protons into the vesicle resulting in gradual endosomal lumen acidification from pH 7.4 to values of about 4.0 in lysosomes [16], where QDs are finally delivered. The possible effect on the QD photophysical properties of the factors listed above, as well as the components of the culture medium necessary for cell growth, has not been adequately studied. However, the assessment of both cytotoxicity and the influence of the microenvironment is extremely important for understanding the prospects for the use of specific QDs in biomedicine and basic research.

In this work, cadmium-free InP/ZnS-based QDs functionalized by PEG with -COOH groups (further indicated as InP-QDs) were the object of study. The choice of this design was based on the fact that untargeted QDs can be chemically bound to streptavidin through -COOH groups of QDs, leaving four biotin-binding sites free. This is one of the most popular approaches to turning untargeted QDs into targeted ones through binding to biotinylated ligands, such as growth factors [17,18]. We address the impact of salts and pH, as well as DMEM culture medium, on the spectral- and time-resolved luminescent characteristics of InP-QDs in in vitro model systems. We also describe InP-QD interaction with cell lines of different origins and perform a single-vesicle analysis of InP-QD luminescence in endosomes. Finally, the concentration-dependent toxicity of InP-QDs is evaluated using two methods.

## 2. Results

### 2.1. Photophysical Characteristics of InP-QDs

#### 2.1.1. Properties of InP-QDs in $H_{2}O$

First, the spectral and luminescent properties of QDs in deionized water were studied (Figure 1). InP-QDs, as is typical for QDs, intensively absorb in the UV range of the spectrum. It was found that the wavelength of 600 nm corresponds to the first exciton absorption band of InP-QDs, with the luminescence maximum at 640 nm (Figure 1a). As no other bands were observed in the luminescence spectrum, the type of luminescence can be attributed to exciton.

The InP core diameter and extinction coefficients of the InP-QDs were calculated using a method designed for CdSe-based QDs by Yu et al. [19]. This is based on the fact that the position of the exciton absorption peak of QDs with a certain organization of electronic levels depends on the semiconductor core size. This estimation of the core diameter is generally accepted. The method was adapted to calculate the sizes of InP-based QDs [20], and it was found that the core diameter of InP-QDs is 4.5 nm and the extinction coefficient is $3.3\times 10^{5}$ M${}^{-1}$ cm${}^{-1}$. It should be noted that this estimation does not consider the thickness of the ZnS layer and the fact that functionalization with PEG typically results in an increase in QD-particle size up to 30–40 nm [21].

The average photoluminescence (PL) lifetime <τ>, estimated from an analysis of the luminescence decay kinetics, is an important photophysical characteristic of nanostructures. It depends on the organization of the electronic levels and is determined not only by radiative, but also by nonradiative transitions. The reported <τ> for InP-based QDs of various structures varies from 20 to 60 ns [22,23]. At the same time, Cd-based QDs have PL lifetimes of about 10–20 ns [24,25].

Figure 1b shows the InP-QD luminescence decay curve in the water and its best-fitting approximation, which uses Equation (Equation 2) (see “Materials and Methods”). The parameters of this approximation are given in Table 1. This analysis shows that the PL lifetime of InP-QDs in our case is approximately 50 ns, which corresponds to the above-cited lifetime for InP-QDs but is significantly higher than that for CdSe/ZnS-based QDs.

According to Equation (Equation 1) described in the “Materials and Methods” section, the calculated quantum yield of luminescence in the water for the InP-QD sample we used was 10–12%, and this estimate did not change for 24 h (Figure 2a). This value is lower than the 31% typical for CdSe/ZnS QDs [26].

#### 2.1.2. Effect of Components of the Physiological Environment on the Photophysical
Characteristics of InP-QDs In Vitro

When QDs are used as fluorescent labels for cells in culture, they first meet growth medium that imitates physiological liquids—buffered water solution with a complex composition of salts, small molecules such as vitamins, and proteins. To maintain cell proliferation, blood serum enriched with growth factors and hormones is usually added. As mentioned above, the effective size of functionalized QDs dictates their entry into the cells via endocytosis. Untargeted QDs enter the cell slowly and proportionally to their concentration in growth medium, while internalization of targeted QDs occurs more effectively, and such QDs undergo rapid concentration in endosomes due to specific molecular mechanisms. Nevertheless, in both cases, QDs are finally delivered to lysosomes [18,25]. There is complex cross-talk between endosomal ion channels and enzymatic activities along the endocytic pathway, resulting in significant regulated changes in ion concentrations [16].

Further, we analyzed the characteristics of InP-QDs in DMEM medium alone and in DMEM supplemented with 10% FBS in vitro (Figure 2a,b). The presence of DMEM and DMEM+FBS did not shift the luminescence maximum of InP-QDs, but significantly decreased luminescence intensity compared to the water-reducing quantum yield by 30–50% in 2 h and by 60% in 24 h. PL lifetime was found to be less sensitive, but it also decreased for about 20% of the control at 2 h after transferring InP-QDs from water to DMEM and reached 37–39 ns (Figure 2b) without further changes up to 24 h.

The nutrient medium DMEM, and especially DMEM+FBS, is close in composition to the extracellular medium, which includes various amino acids and proteins. It is well-known that when nanoparticles enter the biological environment, they interact with proteins to form a nanoparticle–protein complex. This so-called “protein corona” changes the physicochemical properties of nanoparticles, such as size, dispersion, charge, and surface modification of nanoparticles, which leads to the formation of new charge carrier traps. In addition, processes of photoexcitation energy transfer from nanostructures to bound proteins can occur without emitting photons into the medium [27,28]. Besides, the process of aggregation of nanostructures can occur in the biological environment [25]. The aggregation process can be either very fast or very slow. Long-term incubation of nanoparticles in protein-containing solutions can lead to both gradual aggregation and an increase in the size of the protein corona. All these factors lead to a decrease in the quantum yield and lifetime of the luminescence of nanoparticles.

To highlight the possible influence of the main cellular ions on the optical properties of the studied QDs, in the following experiments, we estimated the quantum yield and PL lifetime of InP-QDs in aqueous salt solutions. Very high concentrations (1 M) of NaCl, KCl, and ${\mathrm{CaCl}}_{2}$ were chosen to obtain the maximal effect. We found no effects on QD luminescence spectra in salt solutions (Figure 2). Additionally, as with DMEM, the quantum yield decreased by about 30% compared to the control for all salts checked, but in contrast to the DMEM case, it remained virtually unchanged during 24 h incubation, while the difference of PL lifetimes relative to the values in water was negligible at all time points.

#### 2.1.3. Effect of pH on the Photophysical Characteristics of InP-QDs In Vitro

As mentioned above, internalized QDs localize in the intralumenal space of endosomes, which is acidified during endocytosis due to the operation of the vacuolar proton pump [29,30]. In this regard, we compared the characteristics of QDs in PBS with different pH levels. The PBS standard solution at pH 7.4 is a mixture of several salts, corresponding in osmolarity and ion concentration to physiological fluids (2.7 mM KCl, 137 mM NaCl, 10 mM ${\mathrm{Na}}_{2}{\mathrm{HPO}}_{4}$, and 1.76 mM ${\mathrm{KH}}_{2}{\mathrm{PO}}_{4}$). Acidic pH was adjusted by adding HCl.

The data presented in Figure 3 show that the intensity of InP-QD luminescence, as well as quantum yield and PL lifetime in PBS at neutral pH, is about 7% lower compared to the water, which generally corresponds to the data on the effect of salts. Incubation of InP-QDs in PBS solution with pH 4.0 for 2 h resulted in a decrease in quantum yield of about 25–35%, and incubation for up to 48 h had no additional effect. For the approximation of the decay curves, the amplitudes and components of the QD luminescence lifetime were calculated using Equation (Equation 2), where $A_{i}$ is the amplitude component, and $\tau_{i}$ is the component of the PL lifetime of nanostructures. The average QD luminescence lifetimes <τ> were calculated using Equation (Equation 3). The fact that the best approximation is a multicomponent indicates the complex organization of photoluminescence processes in QD, which is a system of “core/shell and microenvironment” with different contributions ($A_{i}$) of several independent processes with a certain $\tau_{i}$ to the average luminescence lifetime <τ> of the entire ensemble. According to estimations of average PL lifetimes (Table 1), this parameter decreased for 14% from 42 ns at pH 7.4 to 37 ns at pH 4.0. The changes in the InP-QD lifetime in this case, as can be seen, generally correspond to changes in the quantum yield.

### 2.2. InP-QD Interactions with Cells

#### 2.2.1. InP-QD Uptake by Cultured Cells of Different Types

The in vitro data presented above indicate a relatively high level of stability of InP-QDs under physiological-mimicked conditions; therefore, we probed them as cell markers. Experiments were carried out to study InP-QD interactions with macrophage-like cells of line J774, and the cells of two lines of epithelial origin, HeLa and A549, which are incapable of phagocytosis. Using confocal microscopy (Figure 4a), we showed that J774 cells efficiently engulfed InP-QDs in numerous phagosomes. This was expected for professional macrophages, but epithelial cells form significantly smaller endocytic vesicles and, accordingly, are unable to quickly concentrate fluid-phase cargoes, such as untargeted QDs. However, InP-QD luminescence associated with both epithelial cell lines can be detected despite a relatively low quantum yield, and the signal is localized in vesicle-like structures typical of endocytosis. For comparison, in the bottom row of Figure 4, images of the same cell lines incubated with Cd-based nanoparticles of the same design are shown. Note that in HeLa cells, endosomes are smaller and their integral intensity is lower than that in A549, especially for InP-based QDs, possibly due to the lower quantum yield (10% and 31%, respectively). However, even at the low concentration used (20 nM), internalized InP-QDs can be reliably detected. These data are consistent with previously obtained results from other groups with CdSe/ZnS QDs-PEG-COOH [31,32].

To further compare InP-QD behavior in the model cells, we addressed the dynamics of InP-QD uptake in two epithelial cell lines (HeLa, A549) originating from carcinomas, since the study of QD uptake by such cells which are not characterized by active nonspecific consumption of nanoparticles from the external environment is of greater interest in terms of drug delivery to cancer cells. Using flow cytometry, we analyzed the signal distribution of QD luminescence in the cell population (Figure 5a, spectra) and normalized these data per cell (a, insets). It was found that both HeLa and A549 accumulated InP-QDs proportional to the incubation time (Figure 5a) and concentrations (Figure 5b). In addition, A549 internalized InP-QDs faster and more effectively than HeLa, which corresponds to our microscopic data.

#### 2.2.2. Analysis of Photophysical Properties of Cell-Associated InP-QDs

Taking into account the differences found in InP-QD uptake and intracellular localization, the spectral–luminescent properties of InP-QDs in these cell lines were analyzed. Luminescence spectra were taken from InP-QD clusters in the cells incubated with QDs for 24 h, indicated in Figure 6a by arrows. At this time, the untargeted cargo entering the cell by passive endocytosis is accumulated in acidic endolysosomes. The results obtained (Figure 6b) show that the position of InP-QD fluorescence maxima in the cells does not practically deviate from the fluorescence maximum of InP-QDs in aqueous solutions, that indicates the stability of cell-associated InP-QDs.

The microscopic images presented in Figure 4 and Figure 6 show the bright spots of InP-QDs observed in vesicle-like structures. As we analyzed optical sections at the level of the nucleus under the condition of the most highly contrasting DIC images of intracellular structures [25], these vesicular structures are most likely endosomes localized inside the cell. However, it cannot be ruled out that some bright clusters are located on the plasma membrane, that is, outside the cell. To distinguish between internalized and extracellular QD signals, we relied on our experience in identifying endosomes, and only clusters located on the coverslip outside the cells were considered extracellular in our analysis. To understand whether the state of InP-QDs inside and outside the cells differed, we evaluated the decay kinetics of InP-QD luminescence in these two groups. The cells were incubated with InP-QDs (20 nM) for 24 h, and FLIM images were taken (Figure 7).

On the basis of approximations of decay curves recorded by FLIM, the distribution of PL lifetimes over the entire sample was determined with the formation of a colored histogram from blue to red in ascending order (Figure 7a, QDs). Then, to visually separate the QD clusters inside (Figure 7a, QDs Inside) and outside the cells (Figure 7a, QDs Outside) from the overall FLIM image, the indicated areas inside and outside the cells were shown separately.

It can be seen that the luminescence decay curves of the InP-QDs differ in the water, inside and outside the cells (Figure 7b), being lower for internalized QDs compared to QDs outside the cells of both lines. Additionally, our analysis has demonstrated that, regardless of the cell line, outside InP-QD clusters show an average PL lifetime of about 32–38 ns, which is similar to that in nutrient media (37 ns, Figure 2c). The shorter PL lifetime in cells is apparently explained by the fact that QDs get into endolysosomes, where they are concentrated and exposed to an acidic environment.

Importantly, analysis of the data from three independent experiments on at least 50 objects for each variant showed the same clearly reproducible trend, with a large scatter between individual clusters in the same cell (Figure 7c).

#### 2.2.3. Analysis of InP-QD Cytotoxicity

InP-based QDs have been proposed as an alternative to potentially toxic Cd-based QDs. To test this, the cell viability of HeLa and A549 cells was assessed by flow cytometry after incubation with InP-QDs (Figure 8a–c). A cell viability assay using the PI test showed a fairly high level of intact living cells when incubated with InP-QDs at a concentration of 20 nM for both 4 and 24 h (Figure 8a). However, when the concentration of InP-QDs increased to 60 nM, there was a noticeable decrease in the total number of cells (Figure 8b) as well as in the percentage of intact cells (Figure 8c) after incubation with QDs for 24 h. Therefore, we conclude that InP-QDs can be toxic. To prove this by another approach, we estimated the effect of InP-QDs on metabolic activity using an MTT test (Figure 8d) and found that the NADPH-dependent cellular oxidoreductase system is already significantly suppressed by incubation with InP-QDs at 10–20 nM.

It is important to note that when assessing the viability of HeLa and A549 cell lines by fundamentally different methods, a similar picture of cytotoxicity was observed. At the same time, based on the mean values of cell viability according to the PI test (Figure 8c), A549 seems to be more sensitive to increasing concentrations of InP-QDs than HeLa, while the MTT test gives the opposite results (Figure 8d), although the differences are not significant up to a concentration of 60 nM. To understand whether these trends really reflect some differences in the mechanisms of InP toxicity in different cell lines, additional studies are needed.

## 3. Discussion

As photostable bright fluorophores, QDs provide scientists and physicians with the opportunity to study the behavior of internalized labeled proteins and act as an imaging component of theranostic nanoplatforms of various designs. In our study, QDs with an InP/ZnS core functionalized by PEG with -COOH groups were used. Such QDs can be bound to streptavidin, which provides ample opportunities to target them using biotinylated antibodies or ligands. Thus, we have previously shown [17,18] that cadmium-based QDs of the same design form stable complexes with biotinylated Epidermal Growth Factor (EGF) and that, despite a significant increase in the size of the QD-bound ligand, EGF-QD binds to its receptor on the cell surface and passes along exactly the same endocytic pathway as native EGF after internalization. EGF-QDs have proved to be a very useful tool for long-term observations of intracellular processes we have shown, such as fusion and fission of endosomes and transport along microtubules in living cells [33]. Moreover, we have shown that the EGF-QD complex effectively binds to cells overexpressing the EGF receptor, which is often observed in tumor cells of various origins but not in normal cells with low receptor density. This phenomenon is explained by the critical need for the formation of receptor dimers/oligomers for the normal functioning of the EGF receptor system, which is facilitated by the high density of the receptor protein on the cell surface. This property could be used to detect the cells that overexpress the receptor and deliver photosensitizers or anticancer drugs to them. However, it is impossible to completely exclude the interaction of such complexes with normal cells that can be damaged due to the known high toxicity of cadmium and the possibility of the leakage of cadmium ions from nanoparticles. These concerns lead to the search for QDs of another composition that do not contain cadmium, and one such candidate is InP-based QDs. Indeed, many studies have discussed the lower cytotoxicity of In-based QDs compared to Cd-based QDs [34,35,36,37,38], including in vivo research [39].

Our goal was to study in detail the possibility of using InP-QDs, especially in the context of their interactions with cells. First, we showed that InP-QDs have a lower quantum yield and a longer PL lifetime compared to Cd-based QDs of the same design, but they enter the cells in the same way, namely via endocytosis. InP-QDs can be easily detected in endosome-like structures when used at nanomolar concentrations. No reliably detectable signal was seen in the cytosol and nuclei of cells of different origin—macrophage-like J744 cells and two endothelial cell lines of cancer origin (HeLa and A549). Although both epithelial cell lines accumulate InP-QDs with slow dynamic characteristics of fluid-phase endocytosis, the efficiency of accumulation by A549 is significantly higher than by HeLa. This means that the efficiency of QD uptake may vary depending on the endocytic machinery settings of certain cells. The efficiency also does not depend on the composition of the QD core but relies on its surface properties, in particular, on the presence of active groups (-${\mathrm{NH}}_{2}$, -COOH, -OH) and even their number [26,40].

There are practically no data in the literature on the effect of environmental changes on the fluorescence and PL lifetime of In-based QDs, although this issue has been considered for Cd-based QDs and, to a greater extent, from the perspective of pH-dependence. However, any contact with the cell is preceded by contact with physiological solutions, which in experimental conditions are culture media and buffers. As a component of these solutions, large QDs enter the endosomes, where they are met by a constantly increasing concentration of protons ejected by the vacuolar proton pump. First, we addressed the effect of these environmental factors and found that DMEM has the largest impact on InP-QD quenching (PL lifetime decreased by about 20% compared to the water), while high concentrations of ${\mathrm{Na}}^{+}$, $K^{+}$, ${\mathrm{Ca}}^{2+}$ and ${\mathrm{Cl}}^{-}$ have a significantly lower effect on PL lifetime (less than 10%, Figure 2). However, PBS (pH 7.4), which is a more complex mixture of salts in physiological concentrations, reduced the PL lifetime by 15% (Table 1). This indicates the complex nature of the interaction of the medium components with the surface of the quantum dots. In this study, we determined both quantum yield and PL lifetime according to formulas given in the Materials and Methods section, and in all cases, the first parameter demonstrated more dramatic changes than the second. However, as expected, the trends were the same, and in the case of experiments with the cells, PL lifetime was the preferable parameter because it does not depend practically on the concentration of the fluorophore, which can vary from endosome to endosome.

As mentioned earlier, the pH gradient from 7.0–6.8 in early endosomes to 4.5–4.0 in lysosomes is one of the most characteristic features of the endocytic pathway. We have shown that the PL lifetime of InP-QDs was not affected at a pH of 7.4, typical for extracellular media, but decreased by 25% at pH 4.0, which is characteristic of lysosomes. Similar effects have also been reported in other studies with different QDs, although the degree of quenching varied. Thus, Gao et al. reported that almost complete quenching of luminescence occurs upon incubation of mercaptoacetic acid solubilized CdSe/ZnS QDs in PBS at pH 4.0 [41]. Similar results were obtained by another group on InP-QDs coated with 3-mercaptopropionic acid, where a 20% decrease in the InP/ZnS QD luminescence intensity when incubated under similar conditions was observed [37]. We earlier registered 20% and 50% decreases in the intensity of the InP/ZnS QDs-PEG with -COOH and -${\mathrm{NH}}_{2}$ groups, respectively [26]. Mechanisms mediating the effect of acidification are being actively debated, and one of the assumptions is that low pH can damage the QD shell, thus changing the electronic states of nanocrystals. However, the spectral characteristics of InP-QDs both in acidic solution and in endolysosomes do not change compared to the water, that indicates QD stability. Next, quenching occurs immediately upon transferring QDs from a solution with a neutral pH to one with an acidic pH [26] and does not change for at least 48 h in solutions (Figure 2). We assumed that the effect of acidic pH was most likely not associated with severe violations of QD’s core integrity; rather, it is the effect of ionization processes and the emergence or destroying of new bonds on the QD surface that leads to an increase in the probability of nonradiative transitions. Nonetheless, it is important to note that, due to the small size of QDs, the wave functions of an electron and a hole cover the entire volume of nanoparticles and propagate to the surface [42]. With a decrease in the pH level, a partial dissociation of the shell from the QD surface was proposed [43,44,45]; as a consequence, an increase in the contact between the nanocrystal surface and the solution, which is a reservoir for excited electrons from the nanocrystal, led to a higher rate of nonradiative relaxation on the QD surface and a decrease in the photoluminescence intensity.

However, the situation becomes somewhat more complicated when QDs are exposed to protons in endosomes because during endocytosis, they are concentrated in a limited volume of the vesicle. Under such conditions, an increase in aggregation is likely, which also leads to quenching. The degree of aggregation can depend significantly on the nature of the shell. Thus, we have shown by the FLIM approach that alloyed QDs coated with cysteine are very prone to the formation of aggregates [25], while CdSe/ZnS QDs covered with PEG aggregate to a lesser extent [46].

The abovementioned problem of maintaining QD integrity is directly related to the issue of cytotoxicity. It has been proposed that CdSe-QDs probably lose their integrity in the process of entering the cell by endocytosis, resulting in the release of ${\mathrm{Cd}}^{2+}$ ions, which are toxic to cells. At the same time, supposed damage to the integrity of InP-QDs should not be so critical for cell viability due to the fact that ${\mathrm{In}}^{3+}$ ions contribute less to cytotoxic processes. However, our data show that InP/ZnS QDs are rather toxic, and their toxicity grows with the concentrations used. We cannot exclude the fact that under these conditions, more and more ${\mathrm{In}}^{3+}$ ions can escape endosomes and affect some targets in cytosol; however, confocal microscopy methods used are not sensitive enough to detect them. On the other hand, we should consider that the majority of QDs stays in endolysosomes, and the possible mechanisms of cytotoxicity should be related to some specificity of these organelles. Thus, their ability to generate superoxide anion and hydrogen peroxide by NADPH-oxidases of the Nox family, which are associated with the endolysosomal membrane, just into the lumen of endolysosome has not been practically considered. Meanwhile, lysosomes are organelles that actively consume oxygen [47] and, consequently, produce ROS. Indeed, two groups [37,48] showed an increase in ROS production when using InP/ZnS QDs. One way to overcome the problem of partial severing of the QD core as a reason for toxicity (if the problem is only in the core and not in the shell) is to make the ZnS shell more protective by increasing its thickness. This kind of modification should increase the stability of the QD core, which, in turn, should reduce the level of cytotoxicity of QDs. Earlier, we demonstrated that alloyed CdSe/ZnS QDs were not toxic at concentrations up to 200 nM [25]. However, these works and our data indicate that the problem of cytotoxicity should be considered from a broader perspective. It is very probable that QD cytotoxicity is also determined by the type of organic shell and the functional groups on its surface [40,48,49]. In addition, when considering cytotoxicity, it should be taken into account that the same QDs will induce different cytotoxicity on different cell types due to the specificity of a certain cell metabolism [50] and preferable ways of QD entry. We report here that the sensitivity of HeLa and A549 cells to InP-QDs is slightly different but reproducible from experiment to experiment. To understand whether these trends really reflect differences in the energy metabolism settings of cells of different lines, additional studies are needed.

Some researchers [24,51,52] suggest that the acidic lysosomal environment plays a key role in reducing the luminescence of QDs inside cells. However, our study shows that the observed quenching of QD luminescence is an additive effect of several factors, not pH level alone. Moreover, single-vesicle FLIM analysis demonstrates significant homogeneity of the QD PL lifetime from endosome to endosome, even after 24 h of incubation. This suggests that the intra-endosomal microenvironment can vary significantly within a single cell. Further studies will show whether this is due to the existence of different populations of endolysosomes even 24–48 h after incubation of cells with QDs or whether it is explained by the asynchrony of the integrity disruption degree or aggregation of QDs in endosomes.

## 4. Materials and Methods

### 4.1. Analysis of the Spectral-Luminescent Properties of QDs

In this work, we used InP/ZnS-based QDs coated by polyethylene glycol (PEG) with -COOH groups (InP-QDs) with a fluorescence maximum at 640 nm (Mesolight, Suzhou, China). The spectral-luminescent properties of QDs were studied by stationary optical spectroscopy. The absorption and photoluminescence spectra of QD solutions were recorded, respectively, on a UV-3600 spectrophotometer (Shimadzu, Kyoto, Japan) and a Cary Eclipse spectrofluorimeter (Agilent, Mulgrave, Australia). The photoluminescence lifetimes were measured by fluorescence-lifetime imaging microscopy (FLIM) using a MicroTime 100 laser scanning luminescence microscope (PicoQuant, Berlin, Germany). The luminescence decays of the nanostructures in solutions were recorded using a 10× objective and an interference optical filter, transparent at a wavelength of 640 nm, to record the luminescence of InP-QDs. Luminescence was excited using a pulsed diode laser (405 nm). When analyzing the decay of QD luminescence in monolayer cells, luminescence in the range of 430–800 nm was collected using a 100× objective from the area of 80 × 80 microns. FLIM images from these areas were rendered at 512 × 512 pixels. The components of the luminescence decay of these nanostructures were calculated by fitting the luminescence decay curves using Origin 8.5 software.

### 4.2. Estimation of Quantum Yield and Photoluminescence Lifetimes

Usually, the luminescence quantum yield of nanostructures is determined using the comparison method, which involves the use of a sample with a known luminescence quantum yield as a reference. In our case, the fluorescent dye Rhodamine 6G was used as a reference. Thus, to determine the quantum yield of the luminescence of nanostructures, the following equation is used: (1)$$\varphi =\varphi_{0}\times \frac{\int I_{s}d\lambda \times n_{s}^{2}\times D_{ref}}{\int I_{ref}d\lambda \times n_{ref}^{2}\times D_{s}},$$
where $\varphi_{0}$ is the luminescence quantum yield of the reference; $\int I_{s}d\lambda$, $\int I_{ref}d\lambda$ are the integral quantum intensities of the luminescence of the sample and reference, respectively; $D_{s}$, $D_{ref}$ are the optical density at the luminescence excitation wavelength of the sample and reference; and $n_{s}$, $n_{ref}$ are refractive indices of sample and reference solvents.

An average PL lifetime value <τ> is determined from the luminescence multiexponential decay kinetics [53]. To approximate the QD luminescence decay curve, a bi- or tri-exponential dependence (Kohlrausch–Williams–Watt function) was used.
(2)$$I\left(t\right)=\sum_{i}\left(A_{i}\times e^{-\frac{t}{\tau_{i}}}\right),$$
where $A_{i}$ is the amplitude component, and $\tau_{i}$ is the component of the PL lifetime of nanostructures. The best approximation was achieved using a three-exponential model (*i* = 1; 2; 3).

The average PL lifetime is calculated using the equation: (3)$$\langle \tau \rangle =\frac{\sum_{i}A_{i}\times \tau_{i}^{2}}{\sum_{i}A_{i}\times \tau_{i}}.$$

It is known that the PL lifetime time and the quantum yield of the photoluminescence of the fluorophore are related according to the equations [54,55]: (4)$$\varphi_{i}=\frac{\tau_{i}}{\tau_{r}},$$
(5)$$\tau_{i}=\frac{1}{k_{r}+k_{nr}^{i}},$$
where $\varphi_{i}$ is the PL quantum yield of the fluorophore, $\tau_{i}$ is the PL lifetime, $\tau_{r}$ is the radiative time, and $k_{r}$ and $k_{nr}^{i}$ are the radiative and non-radiative rates.

### 4.3. In Vitro Models

To evaluate the influence of most important ions separately, InP-QDs were added to solutions of NaCl, KCl and ${\mathrm{CaCl}}_{2}$ (LenReactiv, St. Petersburg, Russia) prepared on distilled water. Solutions with different pH were prepared on phosphate-buffered saline (PBS, LenReactiv, St. Petersburg, Russia). The required pH level was adjusted with a saturated solution of HCl (LenReactiv, St. Petersburg, Russia). The cell lines used in this work were normally grown in Dulbecco’s modified Eagle growth medium (DMEM; GIBCO, Waltham, MA, USA). The DMEM was based on phosphate buffer (pH 7.4) and contained NaCl, KCl, ${\mathrm{CaCl}}_{2}$ and ${\mathrm{MgSO}}_{4}$·$H_{2}O$, as well as 13 essential amino acids, 6 water-soluble vitamins, choline and inositol acting as a hydrocarbon source, and other compounds (https://www.thermofisher.com/ru/ru/home/technical-resources/media-formulation.8.html; accessed on 30 December 2022). DMEM without the pH indicator Phenol Red was used to avoid unspecific signals.

### 4.4. Cell Lines

The mouse macrophage-like J774 cell line, human cervical adenocarcinoma HeLa cell line, and human lung adenocarcinoma A549 cell line were obtained from the Institute of Cytology, the shared research facility “Vertebrate Cell Culture Collection” supported by the Ministry of Science and Higher Education of the Russian Federation, Agreement No. 075-15-2021-683. The cells were cultured in 25 cm${}^{2}$ flasks or 96-well plates (Nunc, Thermo Scientific, Rochester, NY, USA) in DMEM medium containing 20 mM glutamine, 10% fetal bovine serum and 1% penicillin/streptomycin (GIBCO, Waltham, MA, USA), in the atmosphere of 5% ${\mathrm{CO}}_{2}$ at 37 °C. For experiments, the cells were seeded on coverslips which were placed in Petri dishes in DMEM medium (GIBCO, Waltham, MA, USA), supplemented with 10% fetal bovine serum (FBS; GIBCO, Waltham, MA, USA), and further treatments were performed after 48 h at a 60–70% monolayer. The data from three independent experiments were averaged, and at least 50 cells were taken for every experimental point.

### 4.5. Confocal Microscopy

Cell imaging was performed using an Olympus FV3000 laser scanning confocal microscope (Olympus, Tokyo, Japan) with oil immersion lens 40/1.42×. The fluorescence of QDs was excited by a 405 nm laser and recorded in the range of 600–680 nm. Images were taken in a spectral channel corresponding to the fluorescence detection region of specific QDs and in a channel of differential interference contrast in transmitted light (DIC). When indicated, the fluorescence of QDs was studied in spectral scanning mode (Mode/xyλ) in the range of 520–750 nm with a step of 5 nm.

### 4.6. Analysis of Cell-Associated Fluorescence by Flow Cytometry

Control and incubated with InP-QD monolayer cells were washed three times with PBS and then transferred into suspension by treatment with trypsin solution. Cell-associated InP-QD fluorescence (filter 610/20 BP) was measured using a CytoFLEX cytometer (Beckman Coulter, Brea, CA, USA). The data obtained from 5–10 ×$10^{3}$ cells for each experimental point were analyzed using CytExpert 2.0.0.152 software (Beckman Coulter, Brea, CA, USA).

### 4.7. Estimation of Cell Survival

Living cells have membranes that are still intact and exclude Propidium iodide (PI, Sigma, St. Louis, MO, USA) that easily penetrate the damaged, permeable membranes of non-viable cells. PI at a concentration of 50 μg/mL was added for 1–2 min to the suspensions of control cells and the cells incubated with InP-QDs. Then the cells were analyzed by fluorescence-activated cell-sorting on a CytoFLEX and data on PI fluorescence versus forward-scattering on a logarithmic scale (FSLOG/FL4LOG) were collected as described above. Then the ratio of not-stained living cells versus PI-stained dead cells was calculated.

### 4.8. Evaluation of Cell Metabolic Activity by MTT Test

HeLa and A549 cells were seeded in 96-well plates at 2.5 ×$10^{4}$ cells per well and allowed to grow for 48 h. After that, QDs were added to cells for 24 h at concentrations indicated in the Figure legends. At the end of the incubations, QDs were washed out, the medium was exchanged for fresh DMEM without phenol red (100 µL/well), and then, 10 μL/well of 3-(4,5-dimethylthiazol-2-yl)-2,5-diphenyltetrazolium bromide reagent (MTT, 5 mg/mL, Invitrogen, Eugene, OR, USA) was added. After 4 h of additional incubation, the media were removed, and 50 μL/well of DMSO was added to solve the formed formazan crystals produced by the metabolic active cells, and a Thermo Scientific Multiskan FC microplate format photometer (Thermo Fisher Scientific, Waltham, MA, USA) determining the optical absorption at a wavelength of 570 nm was used to measure the cell viability.

### 4.9. Statistical Data Processing

Statistical data processing was performed using Microsoft Office Excel 2010 (Microsoft Corporation, Albuquerque, NM, USA). The graphs were built using Origin 8.5 (OriginLab, Northampton, MA, USA) software. The bar charts (mean ± standard error of the mean) and box plot were built using Microsoft Office Excel 2010. All results were obtained from at least three independent experiments.

## 5. Conclusions

In this work, InP-based cadmium-free QDs were investigated as a less toxic alternative to CdSe-QDs for biological and biomedical research. InP-QDs with -COOH groups were chosen for analysis because this design allows the binding of strepavidin in the right orientation to engage in further complexing with biotinylated target molecules, such as antibodies or growth factors. Taking into account the fact that QD interaction with the cells is a kind of “two-way road”, we addressed not only QD effect on the cells, such as cytotoxicity, but also possible effects of the intracellular microenvironment on the PL properties of InP-QDs. We showed that despite a lower quantum yield compared to CdSe-QDs, InP-QDs can be reliably detected in the cells, where they enter via endocytosis and accumulate in endolysosomal compartments similar to Cd-based QDs of the same design. The present study indicates that not only can acidification influence InP-QD PL lifetimes, but other components of the intra-endosomal environment can also contribute to total registered changes in QD PL properties. We suggest that these effects are common to all QDs and do not depend on the core composition but mainly on the shell properties. With this in mind, one should be more cautious about the use of QDs as pH sensors. However, InP-QDs can serve as indicators of multifactor changes in intraendosomal composition under different impacts in biomedical research. Unfortunately, we did not confirm the absence of toxic effects of InP-QDs on the cells: two different tests showed significant cell damage at concentrations above 20 nM. The methods to solve this problem are either to strengthen the protective layers of the core (through alloyed QDs, for example) or reduce by several-fold the concentration of QDs used. The latter is possible by creating targeted InP-QDs, thus directing them to internalize via more sensitive and effective receptor-mediated endocytosis, which would provide new opportunities in biomedicine in the future.

## Figures and Tables

**Figure 1 ijms-24-02699-f001:**
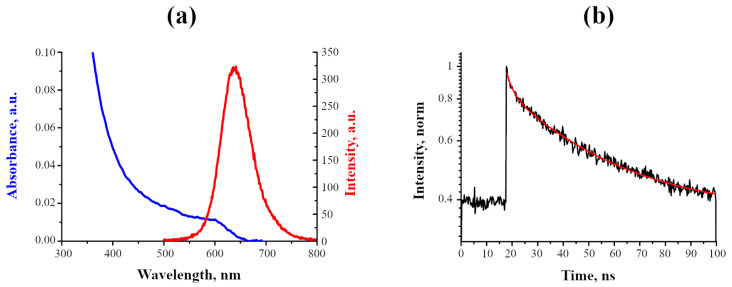
Photophysical properties of InP-QDs (80 nM) in $H_{2}O$. (**a**): Absorption and luminescence spectra; (**b**): Decay curve and its approximation (red line).

**Figure 2 ijms-24-02699-f002:**
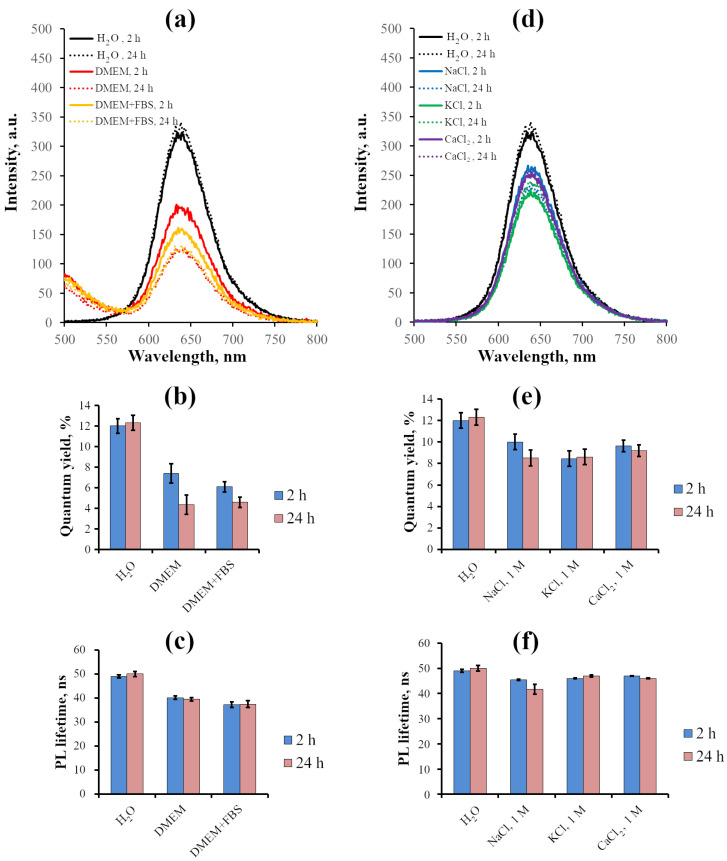
Photophysical characteristics of InP-QDs (80 nM) in various solutions ($H_{2}O$, ${\mathrm{Na}}^{+}$, $K^{+}$, and ${\mathrm{Ca}}^{2+}$ solutions and DMEM) after incubation for 2 and 24 h. (**a**): Luminescence spectra of InP-QDs in DMEM; (**b**): Quantum yields of InP-QD luminescence in DMEM, %; (**c**): Lifetimes of InP-QDs luminescence in DMEM, ns. (**d**): Luminescence spectra of InP-QDs in aqueous salt solutions; (**e**): Quantum yields of InP-QD luminescence in aqueous salt solutions, %; (**f**): Lifetimes of InP-QD luminescence in aqueous salt solutions, ns. Data (**b**,**c**,**e**,**f**) represent the mean ± standard error of the mean from at least three independent experiments.

**Figure 3 ijms-24-02699-f003:**
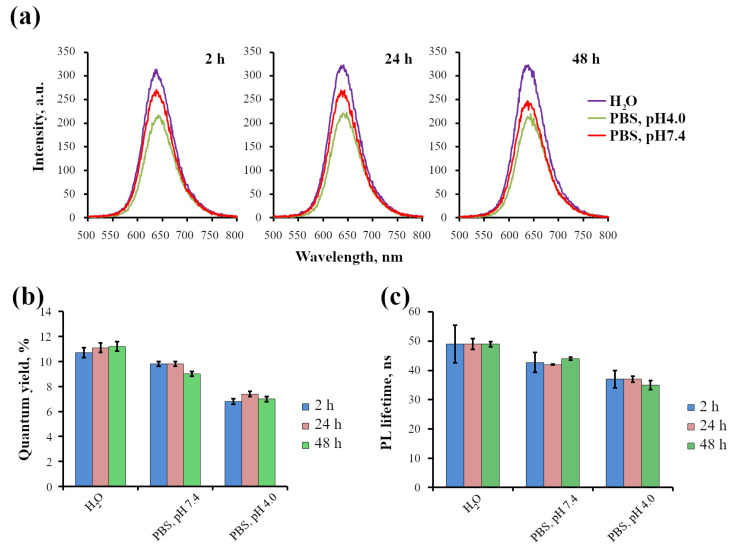
Effect of pH level on the photophysical characteristics of InP-QDs (80 nM). (**a**): Luminescence spectra of InP-QDs in $H_{2}O$ and PBS solutions with different pH after 2, 24, and 48 h of incubation. (**b**): Quantum yields of InP-QD luminescence, %; (**c**): Lifetimes of InP-QD luminescence, ns. Data (**b**,**c**) represent the mean ± standard error of the mean from at least three independent experiments.

**Figure 4 ijms-24-02699-f004:**
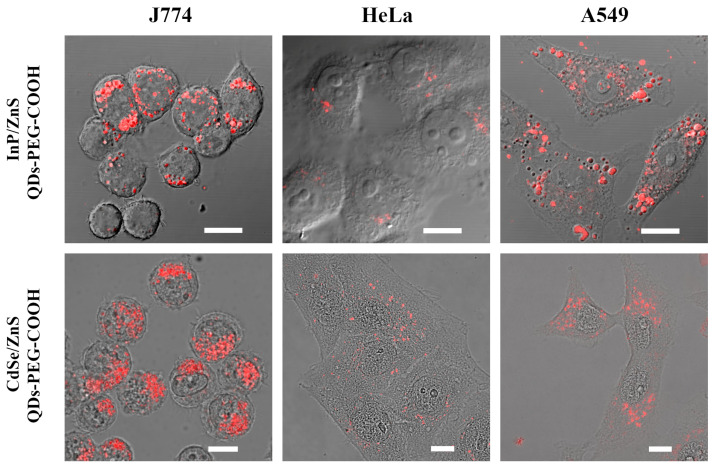
Visualization of InP-QDs and CdSe-QDs in J774, HeLa, and A549 cells after 4 h of incubation with QDs (20 nM). Representative images of the cells are presented. The optical section at the middle of the nucleus was taken and combined with the DIC channel. Scale bars 10 μm.

**Figure 5 ijms-24-02699-f005:**
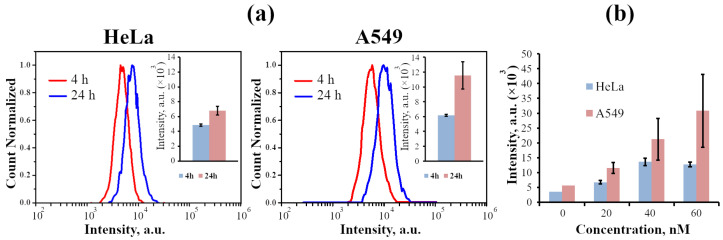
Fluorescence-activated cell sorting analysis of InP-QD uptake by cells. (**a**): Distributions of the fluorescence intensity of the cell population after the incubation of cells with InP-QDs (20 nM) for 4 and 24 h. In the upper right corner, the histograms of the average value of a cell fluorescence intensity after 4 and 24 h of incubation with InP-QDs for HeLa and A549 cells, respectively, are presented. (**b**): Accumulation of InP-QDs depending on the concentration (20, 40, and 60 nM) after 24 h of incubation of cells with InP-QDs. The fluorescence intensity of the control cells without InP-QDs (0 nM) indicates the autofluorescence of the cells.

**Figure 6 ijms-24-02699-f006:**
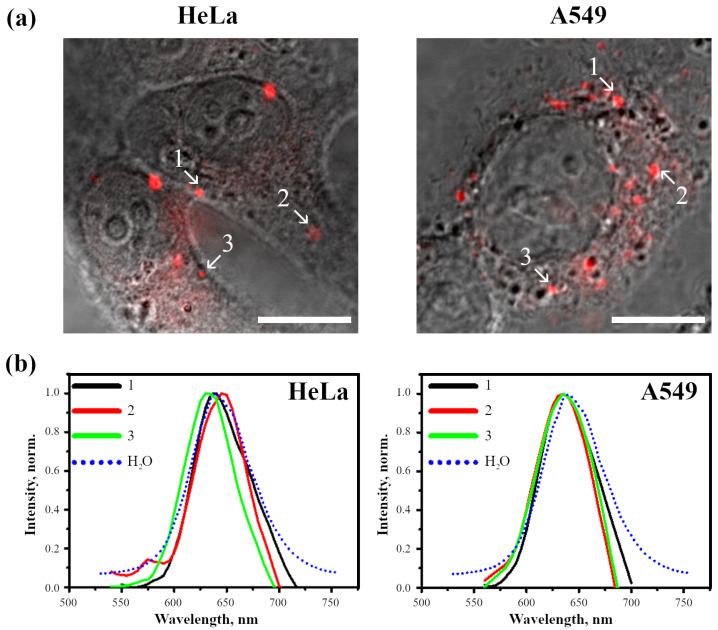
Spectral scanning of InP-QD clusters in cells (20 nM, 24 h) obtained with a confocal microscope and the spectra of the same InP-QDs in $H_{2}O$. (**a**): Confocal images of HeLa and A549 cells. The arrows indicate the selected areas (1–3) for InP-QD spectral scanning. (**b**): Fluorescence spectra of InP-QDs in $H_{2}O$ and in the selected areas (1–3) of the cells in (**a**). For comparison, the spectra were normalized for the maximum intensity value in each case. Scale bars 10 μm.

**Figure 7 ijms-24-02699-f007:**
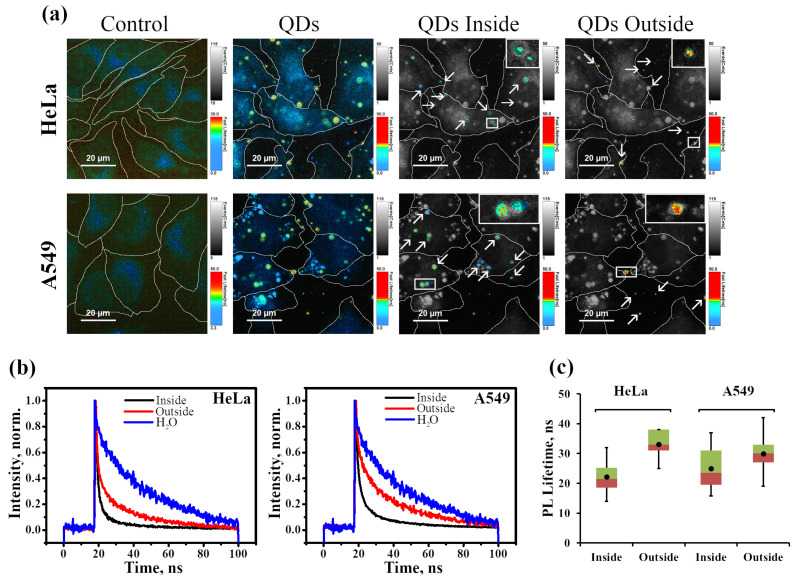
Fluorescence-lifetime imaging microscopy (FLIM) of HeLa and A549 cells incubated with InP-QDs (**a**): Typical FLIM images of control cells and cells after incubation with InP-QDs (20 nM, 24 h). QD-positive clusters inside (endolysosomes) and outside the cells (aggregates) are marked with arrows. Insets represent enlarged views (5×) of the boxed region. (**b**): InP-QD decay curves obtained from FLIM images presented at (**a**) with QDs inside and QDs outside the cells compared to the water. The parameters of these approximations are given in Table 2. (**c**): Statistical analysis of PL lifetimes of 50 individual QD structures per experimental point inside and outside of HeLa and A549 cells from three independent experiments. Data are presented as box plots. In box plots, whiskers represent the minimum and maximum values, bases represent the interquartile range between the first (red) and third quartiles (green), and midlines represent the median. Point symbols indicate the mean value.

**Figure 8 ijms-24-02699-f008:**
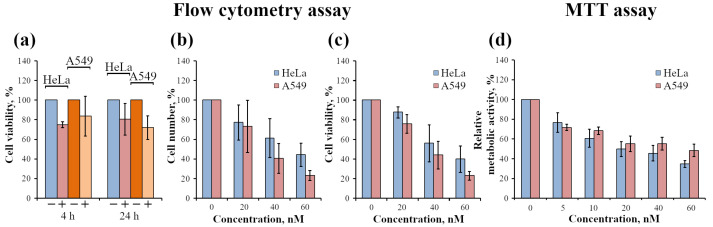
Analysis of the cytotoxic effect of InP-QDs on HeLa and A549 cells by flow cytometry (**a**–**c**) and MTT (**d**) assays. (**a**): Viability (PI test) of control HeLa and A549 cells (–) and after incubation with InP-QDs (20 nM) for 4 and 24 h (+). The proportion of live cells in the control (–) was considered to be 100%. (**b**): Total number of HeLa and A549 cells after incubation of cells with InP-QDs at various concentrations for 24 h (in %). The total number of cells in the control group was considered to be 100%. The changes in the number of cells were estimated separately as a percentage for each line in relation to the corresponding control. (**c**): Viability (PI test) of HeLa and A549 cells during incubation of cells with InP-QDs at various concentrations for 24 h (in % of corresponding control). (**d**): MTT analysis of the metabolic activity of HeLa and A549 cells after 24 h of incubation with InP-QDs. The optical density of formazan produced by the metabolic active control cells not incubated with InP-QDs was taken to be 100%.

**Table 1 ijms-24-02699-t001:** Three-exponential fit parameters of the observed PL lifetime of InP-QDs in $H_{2}O$ and PBS solutions after 24 h of incubation.

	$A_{1}$	$\tau_{1}$, ns	$A_{2}$	$\tau_{2}$, ns	$A_{3}$	$\tau_{3}$, ns	<τ>, ns
** $H_{2}O$ **	0.77 ± 0.012	51 ± 2	0.192 ± 0.02	10 ± 1	0.04 ± 0.01	1.5 ± 1.1	49 ± 2
**PBS, pH 7.4**	0.82 ± 0.01	44 ± 0.27	0.14 ± 0.01	7.6 ± 1.1	0.03 ± 0.01	1 ± 0.65	42 ± 1
**PBS, pH 4.0**	0.79 ± 0.04	38 ± 1.1	0.19 ± 0.02	6.3 ± 0.91	0.02 ± 0.03	0.05 ± 0.04	37 ± 1

**Table 2 ijms-24-02699-t002:** Three-exponential fit parameters of the observed PL decay of InP-QDs interacting with HeLa and A549 cells.

	$A_{1}$	$\tau_{1}$, ns	$A_{2}$	$\tau_{2}$, ns	$A_{3}$	$\tau_{3}$, ns	<τ>, ns
** $H_{2}O$ **	0.77 ± 0.012	51 ± 2	0.192 ± 0.02	10 ± 1	0.04 ± 0.01	1.5 ± 1.1	49 ± 2
**Inside HeLa cells**	0.2 ± 0.02	37 ± 6	0.4 ± 0.1	7 ± 2	0.4 ± 0.1	2 ± 0.3	26 ± 5
**Outside HeLa cells**	0.45 ± 0.001	44 ± 4	0.39 ± 0.002	9 ± 1	0.16 ± 0.004	2 ± 0.4	38 ± 4
**Inside A549 cells**	0.25 ± 0.002	29 ± 3	0.44 ± 0.003	5 ± 0.7	0.31 ± 0.005	1.4 ± 0.2	22 ± 2
**Outside A549 cells**	0.57 ± 0.005	35 ± 2	0.38 ± 0.001	6 ± 0.5	0.1 ± 0.005	2 ± 1	32 ± 1

## Data Availability

The data presented in this study are available from the authors upon request.

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
