# Peer review of "Microenvironmental Impact on InP/ZnS-Based Quantum Dots in In Vitro Models and in Living Cells: Spectrally- and Time-Resolved Luminescence Analysis"

_ijms, 2023, doi:10.3390/ijms24032699_

Round 1

Reviewer 1 Report

The authors reported on the optical properties InP/ZnS-based quantum dots (QDs) under different environmental conditions in vitro models.

InP/ZnS QDs are emerging in biological technologies for fluorescent imaging. The study of this work is systematic. However, the English writing of this manuscript should be greatly improved before re-submission and I suggest the authors letting native English speakers to revise the manuscript again in terms of the English writing sytle. All in all, I recommend the publication of this manuscript upon the following conditions are well addressed.

[1] There are lot of abbreviations not mentioned before such as “DMEM”, “FLIM”, “FBS” and “FACS”.

[2] There are many typos and grammatical mistakes in this manuscript, particularly in the abstract. For example, “In was demonstrated……” and “Despite …… were…….” and “The single-vesicle FLIM analysis…… show……”.

[3] In Table 1&2, the authors did not explain clearly what are A1–A3 and τ1– τ3 and their corresponding physical meaning? And how the average τ was obtained?

[4] In Figure 5b, how did the fluorescence intensity come from the control incubation without InP-QDs?

[5] In Figure 7c, the authors did not mention clearly what are the dots, green and red portions representing for in the figure.

Author Response

We are very grateful for your valuable comments and appreciate your inputs that definitely improve our manuscript.

Below are our point-by-point responses:

Point 1: There are lot of abbreviations not mentioned before such as “DMEM”, “FLIM”, “FBS” and “FACS”.

Response 1: We remove the abbreviations from the Abstract and introduce them in the text of the article upon first use, as well as in “Abbreviations ” section:

FLIM (Fluorescence-lifetime imaging microscopy) was decoded on line 72 (new version 79);

DMEM (Dulbecco’s modified Eagle medium) was decoded on line 109 (new version 117);

FBS (Fetal Bovine Serum) was decoded on line 124 (new version 134);

Since the abbreviation FACS was used in the article only few times, we  have changed it for the full name of the method “Fluorescence activated cell sorting” (line 147 (new version 134) and figure caption 5).

Point 2: There are many typos and grammatical mistakes in this manuscript, particularly in the abstract. For example, “In was demonstrated……” and “Despite …… were…….” and “The single-vesicle FLIM analysis…… show……”.

Response 2: To improve the language we used the proofreading services at https://www.scribbr.com/.

Point 3: In Table 1&2, the authors did not explain clearly what are A1–A3 and τ1– τ3 and their corresponding physical meaning? And how the average τ was obtained?

Response 3: Thank you for this significant comment, it is really a question we have asked ourselves as biologists.

In optics, it is customary to describe decay curves using a combination of mathematical functions (exponential in this case), which gives the maximum agreement with the experimental curve, and the formulas according to which the parameters of these functions are calculated are given in Materials and Methods (equations (2)). Our curves are best described by a combination of three functions, and as biologists we were of course very interested in the very question you asked. Discussing it with our fellow physicists, we realized that even opticians have slightly different interpretations. However, simplifying, we can say that when our fluorophore (QD) is excited, at least three independent photoluminescent processes are stimulated, each of which has its own decay time (τi) and a certain fluorescence intensity at the initial stage of excitation. The last parameter is therefore called the amplitude (Ai). Accordingly, the average decay time is calculated (equations (3)) on the basis of the luminescence lifetimes of all three components, taking into account their contribution (which, in fact, is Ai). In general, the more complex the electronic organization of the fluorophore, the more independent emission processes can be realized. Unfortunately, nothing can be said a priori about the specific process described by each component, and serious physical experiments are required to identify them. The absolute value of the amplitude also does not have a specific meaning in itself, but when compared with the amplitudes of other components, it gives an idea of which process has a greater contribution to the average decay time of the entire system. This, obviously, is not within the scope of this article, and we are not specialists capable of solving it. Moreover, a detailed physical and chemical analysis can reveal a greater number of processes, especially in such complex systems as QDs, however, when modeling, the description of processes is simplified to reasonable levels. Since, as mentioned above, we are not great specialists in the issue discussed in the commentary, and so we would like to limit ourselves to such an addition to the text (line 240, (new version 262)):

"For the approximation of the decay curves, the amplitudes and components of the QD luminescence lifetime were calculated using equation (2), where Ai is the amplitude component, and τi is the component of the PL lifetime of nanostructures. The average QD luminescence lifetimes <τ> were calculated using equation (3). The fact that the best approximation is multicomponent indicates the complex organization of photoluminescence processes in QD, which is a system of "crystalline core-shell-microenvironment" with different contributions (Ai) of several independent processes with a certain τi to the average luminescence lifetime <τ> of the entire ensemble."

Point 4: In Figure 5b, how did the fluorescence intensity come from the control incubation without InP-QDs?

Response 4:  Cells contain a fairly significant number of different molecules that have the property of photoluminescence in a wide spectrum of wavelengths, mainly in the blue-green region, but also in the range of longer wavelengths. This background is known as "autofluorescence" and is recorded even in the absence of internalized fluorophores. We agree that an explanation should be added to Figure 5:

“The fluorescence intensity of the control cells without InP-QDs (0 nM) reflects the autofluorescence of the cells.”

Point 5: In Figure 7c, the authors did not mention clearly what are the dots, green and red portions representing for in the figure.

Response 5: You're right, indeed, there is no consensus on exactly how to build box-and-whisker charts, so this information must be given in the text. So we added a description to Figure 7:

“In the box plots, the whiskers represent the minimum and maximum values; boxes represent the interquartile range between the first (red) and third quartiles (green); and the middle lines represent the median. Dot symbols indicate the mean value."

Reviewer 2 Report

Comments to the authors

The manuscript entitled “Microenvironmental impact on InP/ZnS-based Quantum Dots in in vitro models and in living cells: sprectrally- and time-resolved luminescence analysis” by Litvinov et al. investigated whether core/shell InP/ZnS QDs (InP-QDs) may be an alternative to toxic Cd-based QDs. Generally, the topic of the research is meaningful, and the workload is relatively in place. However, there are many concerns need to be addressed, for example, the logic of this article is not clear enough, the results are not detailed enough, which need carefully revised.

Some specific concerns:

1. Line 21, please do not ignore the logical relationships and should clarify the logic between quantum dots and nonradiative recombination the core.

2. The manuscript needs to cite more related references for support. In Line 178, “the fact” is used for CdSe-based quantum dots and more articles are needed to support whether it can be applied to InP-QDs and its generality.

3. In 3.2.1, the untaken of InP-QDs by J774, HeLa and A549 was studied. However, only HeLa and A549 were studied later. The reason of Line 263-269 is not sufficient and more reasons are needed to elaborate why only HeLa and A549 were studied subsequently.

4. The format of the manuscript as well as the figures should be carefully checked, e.g.

Figure 8 (d) is clearly different from (b) (c). And there are also problems with the use of punctuation in some of the sentences in the manuscript.

5. Figure 8 (c), the cell viability of A549 is not 100%.

Reviewer 3 Report

The authors reported an interesting work synthesis of  core/shell InP/ZnS QDs and used them for biological and biomedical studies, such as single-vesicle analysis in endosomes. This paper is well written, and the work is also well done. However, there are some flaws in the data, some references are missing, some control experiments needed, and some further discussion about data is necessary. Therefore, I suggest a major revision, and following questions must be addressed. After that, I will reconsider the manuscript for publication.

1. The full name of “DMEM” should be presented at the first time the shown in the manuscript.

2. Figure 2a, the PL intensity decreased a lot with prolonged time in DMEM. Why? Any hypothesis and any evidence to support the hypothesis? Please add some discussion in the main text.

3. To show a bigger picture to the readers, the introduction about quantum dots could be expanded a bit and more applications and advantages of QDs need to be discussed. The authors should show why they used PbS QDs instead of nanoparticles, nanowires etc. The following recent example of QDs applications must be cited (https://doi.org/10.1039/C9PY01604J; https://doi.org/10.1021/acsmacrolett.9b00891; https://doi.org/10.1063/5.0051893).

4. For the cytotoxicity testing, the authors believed it is because of In3+ leaking. However, is that also possibly due to the Zn2+? Is that possible to perform ICP testing to prove this point?

5. Figure 1a should be improved. The absorption and PL curve used the same color, as well as the y axis at left and right, which make it unclear for readers to understand. Please revise it, use arrows or different color to present the data.

Author Response

We are very grateful for your valuable comments and appreciate your inputs that definitely help to improve our manuscript.

Below are our point-by-point responses to the Reviewer’s comments:

Point 1: The full name of “DMEM” should be presented at the first time the shown in the manuscript.

Response 1: Thank you for bringing this matter to attention, in this regard, we removed the abbreviations from Abstract and introduced them in the text of the article.

DMEM (Dulbecco’s modified Eagle medium) decoded on line 109 (new version 117).

Point 2: Figure 2a, the PL intensity decreased a lot with prolonged time in DMEM. Why? Any hypothesis and any evidence to support the hypothesis? Please add some discussion in the main text.

Response 2: Your remark is very important, in this regard we have added a description on line 218 (new version 230):

“The nutrient medium DMEM, and especially DMEM+FBS, is close in composition to the extracellular medium, which includes various amino acids and proteins. It is well known that when nanoparticles enter the biological environment, they interact with proteins to form a nanoparticle-protein complex. This so-called "protein corona" changes the physicochemical properties of nanoparticles, such as size, dispersion, charge, surface modification of nanoparticles, which leads to the formation of new charge carrier traps. In addition, processes of photoexcitation energy transfer from nanostructures to bound proteins can occur without emitting photons into the medium (Abraham et al., 2011; Qu et al., 2020). Besides, the process of aggregation of nanostructures can occur in the biological environment (Matyushkina et al., 2022). The aggregation process can be either very fast and very slow. Long-term incubation of nanoparticles in protein-containing solutions can lead to both gradual aggregation and an increase in the size of the protein crown. All these factors lead to a decrease in the quantum yield and lifetime of the luminescence of nanoparticles. ".

Abraham B.G., Tkachenko N.V., Santala V., Lemmetijnen H. and Karp M. (2011) Fluorescence Resonance Bidirectional Energy Transfer (FRET) in Mutated and Chemically Modified Yellow Fluorescent Protein (YFP) Bioconjugate Chemistry, 22(2), 227–234 doi:10.1021/bc100372u.

Qu, S., Sun, F., Qiao, Z., Li, J., & Shang, L. (2020) .In Situ Investigation on Quantum Dot Protein Crown Formation Using Fluorescence Resonant Energy Transfer, Small, 1907633. doi:10.1002/smll.201907633.

Matyushkina A., Litvinov I., Bazhenova A., Belyaeva T., Dubavik A., Veniaminov A., Maslov V., Kornilova E., Orlova G. A. Research time- and spectral-resolved photoluminescence of doped CdxZn1-xSeyS1-y/ZnS quantum dots and their nanocomposites with SPION in living cells, Int. J. Mol. Sci., 2022, 23(7), 4061.

Point 3: To show a bigger picture to the readers, the introduction about quantum dots could be expanded a bit and more applications and advantages of QDs need to be discussed. The authors should show why they used PbS QDs instead of nanoparticles, nanowires etc. The following recent example of QDs applications must be cited (https://doi.org/10.1039/C9PY01604J; https://doi.org/10.1021/acsmacrolett.9b00891; https://doi.org/10.1063/5.0051893).

Response 3: Please take into account that the aim of the article was not so much to study the shape or composition of nanoparticles, but rather to compare quantum dots based on InP with quantum dots based on CdSe that are considered toxic. This article is in line with our research of the interaction of QDs with cells for possible applications in basic research and medicine. In the 1990s, a lot of research was done with nanostructures of various shapes, and the main data indicated that round nanoparticles themselves are less toxic than more rigid nanorods, etc. We focus on quantum dots by the fact that they are more similar to natural globular proteins and thus, we can expect a smaller damaging potential that is not related to their chemical nature but just to the form.

However, you are quite right that the area of QD application can be much wider and the references you give are extremely interesting and therefore we have included them as refs in the Introduction with appropriate comment on line 28 (new version 30):

“Currently, new fields of QDs applications are being developed, in particular, an interesting example is the use of QDs as a photocatalyst for the polymerization of monomers of different origins. From the biomedicine point of view, it seems particularly important to suggest that complexes with QDs can be created in this way to obtain targeted nanoparticles (McClelland et al., 2019, Zhu et al., 2020, Zhu et al, 2021).”

McClelland, K. P.; Clemons, T. D.; Stupp, S. I.; Weiss, E. A. Semiconductor Quantum Dots Are Efficient and Recyclable Photocatalysts for Aqueous PET-RAFT Polymerization. ACS Macro Letters 2019, 9(1), 7—13.

Zhu, Y.; Egap, E. PET-RAFT polymerization catalyzed by cadmium selenide quantum dots (QDs): Grafting-from QDs photocatalysts to make polymer nanocomposites. Polymer Chemistry 2020, 11, 1018–1024.

Zhu, Y.; Jin, T.; Lian, T.; Egap, E. Enhancing the efficiency of semiconducting quantum dot photocatalyzed atom transfer radical polymerization by ligand shell engineering. The Journal of Chemical Physics 2021, 154(20), 204903.

Point 4: For the cytotoxicity testing, the authors believed it is because of In3+ leaking. However, is that also possibly due to the Zn2+? Is that possible to perform ICP testing to prove this point?

Response 4: Actually, ZnS shell not only improves the spectral and optical characteristics of QDs, but also reduces toxicity, and this was shown in several studies. For example, when we used alloyed CdSe-based QDs with thiсk layer of ZnS we did not observe significant toxicity (Matiushkina   et al., 2022). It was shown that coating the core of QDs with a double-thickness ZnS shell in comparison with one-layer reduce the cytotoxicity of QDs (Chibli et al., 2011). Unfortunately, we have no opportunity to perform ICP testing, but with the greatest probability such leakings could not be registered, since Zn is a vital trace element involved as a co-factor in the regulation of the normal functioning of many cellular proteins and processes, and is contained in cells in sufficient quantities. Zinc is also present in commercial multivitamin formulations. Sure, like any other vital substance, it can become toxic at micromolar concentrations (Adhikari et al., 2015), but we use QD at low nanomolar concentrations.

In fact, we are closer to the idea expressed in the same article of Chlibli et al., concerning possible provocation by quantum dots of an increase in the generation of reactive oxygen species, the possibility of which has been demonstrated in the literature and is discussed in the Discussion paragraph. In our study we have limited ourselves to standard definitions of cytotoxicity. The nature of cytotoxicity may be very complex and should be the goal of special detailed study.

Adhikari, S.; Lamichhane, B.; Shrestha, P.; Shrestha B.G. Effect of Invitro Zinc (II) supplementation on Normal and Cancer Cell lines, IJRASET, 2015, Volume 3 Issue I, pp 233-240.

Chibli, H.; Carlini, L.; Park, S.; Dimitrijevic, N.M.; Nadeau, J.L. Cytotoxicity of InP/ZnS quantum dots related to reactive oxygen species generation. Nanoscale. 2011 , 3(6), 2552.

Point 5: Figure 1a should be improved. The absorption and PL curve used the same color, as well as the y axis at left and right, which make it unclear for readers to understand. Please revise it, use arrows or different color to present the data.

Response 5: We are grateful for the advice - indeed, in the version you proposed, Figure 1 looks much clearer. We marked the absorption and luminescence spectra and their corresponding axis labels with different colors.

Round 2

Reviewer 3 Report

The revised version improved a lot and the authors revised the manuscript with care. I suggest publishing this excellent paper in current form.